# Prediction of Long-Term Stroke Recurrence Using Machine Learning Models

**DOI:** 10.3390/jcm10061286

**Published:** 2021-03-20

**Authors:** Vida Abedi, Venkatesh Avula, Durgesh Chaudhary, Shima Shahjouei, Ayesha Khan, Christoph J Griessenauer, Jiang Li, Ramin Zand

**Affiliations:** 1Department of Molecular and Functional Genomics, Geisinger Health System, Danville, PA 17822, USA; vidaabedi@gmail.com (V.A.); vavula1@geisinger.edu (V.A.); jli@geisinger.edu (J.L.); 2Biocomplexity Institute, Virginia Tech, Blacksburg, VA 24061, USA; 3Geisinger Neuroscience Institute, Geisinger Health System, Danville, PA 17822, USA; dpchaudhary@geisinger.edu (D.C.); sshahjouei@geisinger.edu (S.S.); akhan2@geisinger.edu (A.K.); cgriessenauer@geisinger.edu (C.J.G.); 4Research Institute of Neurointervention, Paracelsus Medical University, 5020 Salzburg, Austria

**Keywords:** healthcare, artificial intelligence, machine learning, interpretable machine learning, explainable machine learning, ischemic stroke, clinical decision support system, electronic health record, outcome prediction, recurrent stroke

## Abstract

Background: The long-term risk of recurrent ischemic stroke, estimated to be between 17% and 30%, cannot be reliably assessed at an individual level. Our goal was to study whether machine-learning can be trained to predict stroke recurrence and identify key clinical variables and assess whether performance metrics can be optimized. Methods: We used patient-level data from electronic health records, six interpretable algorithms (Logistic Regression, Extreme Gradient Boosting, Gradient Boosting Machine, Random Forest, Support Vector Machine, Decision Tree), four feature selection strategies, five prediction windows, and two sampling strategies to develop 288 models for up to 5-year stroke recurrence prediction. We further identified important clinical features and different optimization strategies. Results: We included 2091 ischemic stroke patients. Model area under the receiver operating characteristic (AUROC) curve was stable for prediction windows of 1, 2, 3, 4, and 5 years, with the highest score for the 1-year (0.79) and the lowest score for the 5-year prediction window (0.69). A total of 21 (7%) models reached an AUROC above 0.73 while 110 (38%) models reached an AUROC greater than 0.7. Among the 53 features analyzed, age, body mass index, and laboratory-based features (such as high-density lipoprotein, hemoglobin A1c, and creatinine) had the highest overall importance scores. The balance between specificity and sensitivity improved through sampling strategies. Conclusion: All of the selected six algorithms could be trained to predict the long-term stroke recurrence and laboratory-based variables were highly associated with stroke recurrence. The latter could be targeted for personalized interventions. Model performance metrics could be optimized, and models can be implemented in the same healthcare system as intelligent decision support for targeted intervention.

## 1. Introduction

Predictive modeling of stroke, the leading cause of death and long-term disability [1], is crucial due to high individual and societal impact. Each year, about 800,000 people experience a new or recurrent stroke in the United States [2]. It has been estimated that the 5-year risk of stroke recurrence is between 17% and 30% [3,4]. Recurrent stroke has a higher rate of death and disability [5]. Therefore, the identification of patients who are at a higher risk of recurrence can help the care-providers prioritize and define more vigorous secondary prevention plans for those at risk, especially when there are limited resources.

To date several predictive models of recurrent stroke, using regression or other statistical methods, have been developed; however, the clinical utility of these models tends to be limited due to the narrow scope of variables used in these models [6]. In a recent study, multivariable logistic models of 1-year stroke recurrence, developed based on 332 patients, using clinical and retinal characteristics (using 20 variables) have shown promising results with an area under the receiver operating characteristic (AUROC) curve of 0.71–0.74 [7]. Large real-world patient-level data from electronic health records (EHR) and machine learning (ML) methods can be leveraged to capture a greater number of features to help build better prediction models [8]. In a recent study of 2604 patients, ML has been successfully used to predict the favorable outcome following an acute stroke at three months [9]. We also showed that ML can be used for flagging stroke patients in the emergency setting [10,11,12].

The present study aimed at using rich longitudinal data from EHR to construct an ML-enabled model of long-term (up to 5-years) recurrent stroke. We evaluated Extreme Gradient Boosting (XGBoost), Gradient Boosting Machine (GBM), Random Forest (RF), Support Vector Machine (SVM), and Decision Tree (DT), and benchmarked these algorithms’ performance against Logistic Regression (LR) as these are interpretable models and feature importance can be extracted for further validation and assessment by care providers. We hypothesized that (1) all of the modeling algorithms can be trained to predict long-term stroke recurrence, (2) A wide range of clinical features associated with stroke recurrence can be identified, and (3) performance metrics can be improved through sampling processes.

## 2. Methods

All of the relevant codes developed as well as summary data generated for this project can be found at https://github.com/TheDecodeLab/GNSIS_v1.0/tree/master/ModelingStrokeRecurrence (accessed on 19 March 2021).

### 2.1. Data Source

Database description and processing: this study was based on the extracted data from the Geisinger EHR system, Geisinger Quality database, and the Social Security Death database to build a stroke database called “Geisinger Neuroscience Ischemic Stroke (GNSIS)” [13]. GNSIS includes demographic, clinical, laboratory data from ischemic stroke patients from September 2003 to May 2019. The study was reviewed and approved by the Geisinger Institutional Review Board to meet “non-human subject research”, for using de-identified information.

The GNSIS database was created based on a high-fidelity and data-driven phenotype definition for ischemic stroke developed by our team. The patients were included if they had a primary hospital discharge diagnosis of ischemic stroke; a brain magnetic resonance imaging (MRI) during the same encounter to confirm the diagnosis; and, an overnight stay in the hospital. The diagnoses were based on International Classification of Diseases, Ninth/Tenth Revision, Clinical Modification (ICD-9-CM/ICD-10-CM) codes. For each index stroke, the following data elements were recorded: (1) date of the event, (2) age of the patient at the index stroke, (3) encounter type, (4) ICD code and corresponding primary diagnosis of index stroke, (5) presence or absence (and date) of recurrent stroke, and (6) ICD code and corresponding primary diagnosis for the recurrent stroke. Other data elements include sex, birth date, death date, last medical visit within the Geisinger system, presence or absence of comorbidities, presence or absence of a family history of heart disorders or stroke, and smoking status. In the case of multiple encounters due to recurrent cerebral infarcts, the first hospital encounter was considered as the index (first-time) stroke. To improve the accuracy of comorbidity information based on ICD-9-CM/ICD-10-CM diagnosis, either two outpatient visits or one in-patient visit were used to assign a diagnosis code to a patient. Our database interfaces with the Social Security Death Index on a biweekly basis to reflect updated information on the vital status. The manual validation of a random set of patients, including reviewing the MRI, to ensure all patients in the GNSIS database had a correct diagnosis of acute ischemic stroke indicated a specificity of 100%.

Data pre-processing: Units were verified and reconciled if needed and distributions of variables were assessed over time to ensure data stability. The range for the variables was defined according to expert knowledge and available literature—and outliers were assessed and removed. To ensure that patients were active, the last encounter of patients was recorded.

### 2.2. Study Population

For this study, we excluded patients with recurrent stroke within 24 days of the index stroke. We organized the included patients into six groups. One control group and five case groups. The control group consisted of patients who did not have a stroke recurrence during the 5-year follow-up. Case groups 1, 2, 3, 4, and 5 comprised of patients who had a recurrent stroke between 24 days and 1, 2, 3, 4, and 5-years, respectively. The 24 day cut-off was selected to ensure that the recurrent stroke was independent of the index stroke; as our data demonstrate, the number of stroke recurrences stabilizes after approximately 24 days (Figure 1A). Nevertheless, we repeated the analysis by including the patients with a stroke recurrence within the 24 days for comparison. Patients with stroke-related or other vascular death might be excluded from this study if they did not meet the inclusion/exclusion criteria stated above.

### 2.3. Data Processing, Feature Extraction, and Sampling

Training-testing set: Each of the cases and control groups was randomly split into 80:20 training and testing sets.

Imputation: A total of 53 features were used. Table 1 includes data on the missingness. Imputation of the missing values was performed separately on training and testing sets using Multivariate Imputation by Chained Equations (MICE) package [14]. The quality of the imputations was examined using t-test, summary statistics, as well as strip and density plots of the missing features to ensure distribution of the variables was comparable before and after imputation. Only four variables suffered from missingness at relatively higher levels.

Feature selection: We performed feature selection using different strategies. The feature sets were: Set 1: all features; Set 2: all features except medication history; Set 3: features selected by at least two data-driven strategies; and Set 4: minimum set, obtained as the intersect of Set 2 and Set 3 (Appendix A). The full set of features (Sets 1, 2) were selected based on clinical expertise and previous studies [6,15]. Feature selection (Sets 3, 4) was performed based on three data-driven approaches for each set of case-control.

The data-driven approaches were: (1) filter-based methods including Pearson correlation [16] and univariate filtering; (2) embedded methods including RF [17] and Lasso Regression [18]; and (3) wrapper methods including the Boruta algorithm [19] and recursive feature elimination. Feature importance scores were scaled between zero and 100, with higher scores representing higher variable contributions. Using the reduced set of features will ensure variables with high collinearity are removed.

Sampling: The training dataset after applying the case-control definition was imbalanced. Many of the classification models trained on class-imbalanced data are biased towards the majority class. To avoid poor performance of minority class (recurrent stroke) compared with the dominant class, we balanced out the number of cases and controls by up-sampling and down-sampling methods. We applied the up-sampling method to the prediction window with the lowest and median rate of stroke recurrence and down-sampling to the prediction window with the median rate of stroke recurrence. In the up-sampling, we used the Synthetic Minority Over-sampling Technique (SMOTE) [19]. In the down-sampling, we randomly selected patients from the control group.

### 2.4. Model Development and Testing

We used six interpretable ML algorithms and four feature sets to develop a classification model for 1, 2, 3, 4, and 5-year recurrence prediction window. We developed 24 models for each prediction window. The ML algorithms included LR [20], XGBoost [21], GBM [22], RF [17], SVM [23], and DT [24]. We included SVM, LR, and DT as these could provide benchmarking metrics as well as better flexibility in terms of implementation into cloud-based EHR vendors. Therefore, simpler and faster models could provide strategic alternatives for future implementation if the results from this study indicate, similar to other studies [25], that by including a large number of features, models can reach convergence to the point of algorithm indifference (or marginal improvements). A parameter grid was built to train the model with 10-fold repeated CV with 10 repeats. Furthermore, 5-fold repeated CV for the prediction window with the median rate of stroke recurrence was also performed. Model tuning was performed by an automatic grid search with 10 different values to try for each algorithm parameter randomly. For each model, we used 20% of the data for model testing and calculated specificity, recall (sensitivity), precision (positive predictive value, PPV), AUROC, F1 score, accuracy, and computation time for model training. The modeling pipeline is summarized in Figure 1B.

## 3. Results

All of the detailed summary results with comprehensive performance metrics, feature importance and computation time for the 288 models this project are provided as Appendix A.

### 3.1. Patient Population and Characteristics

A total of 2091 adult patients met the inclusion criteria; 114 patients had a recurrent stroke within 24 days from their index stroke and were excluded from the analysis (Figure 1A). Out of 2091 patients, 51.6% were men. The median age was 68.1 years (IQR (interquartile range) = 58–77). The three most common comorbidities were hypertension (72%), dyslipidemia (62%), and diabetes (29%). Table 1 includes the patients’ demographics and past medical history. The rate of stroke recurrence was 11%, 16%, 18%, 20%, and 21% at 1, 2, 3, 4, and 5-year window, respectively.

This study was based on 53 features. Appendix A summarizes the results from the feature selection process. Age, sex, BMI, systolic blood pressure, hemoglobin, high-density lipoprotein (HDL), creatinine, smoking status, chronic heart failure, chronic kidney disease, diabetes, hypertension, and peripheral vascular disease were selected by all of the different data-driven approaches for the five different case-control designs.

### 3.2. Models Can Be Trained to Predict the Long-Term Stroke Recurrence

Model AUROC was stable for the five case-control designs with the highest score for the 1-year prediction window and the lowest score for the 5-year window (Figure 2, Appendix A). The best AUROC for the 1-year prediction window was 0.79 (Appendix A, model#63). The top ten models (AUROC: 0.79–0.74) were from the 1-year prediction window. The best AUROC for the 2, 3, 4 and 5-year prediction windows were 0.70, 0.73, 0.73, and 0.69 respectively. Furthermore, when comparing features included in the models, the AUROC was highest when all of the features were used. The variation in AUROC was higher across the various study window and feature sets for DT, while the score variance was lowest for RF. The ROC curve for the different models is shown in Figure 3 for the 1-year prediction window.

Based on the accuracy, RF (RF, mtry = 14) model, using 26 features (Set3), had the best performance for a 1-year prediction window (accuracy: 90% (95% CI: 86%–92%), PPV: 80%, specificity: 100%). The average accuracy by using the six models and four sets of features was 88% (Appendix A, model numbers 1–24). The prediction accuracy decreased as the prediction window widened to 2-years (average accuracy: 85%) with the best accuracy score reached by LR (86%, 95% CI: 82%–89%) and PPV of 80% with a specificity of 99%, Appendix A model number 79. The average accuracy of the 3-year prediction window was 82% for the 4-year prediction window. The average accuracy of the 5-year prediction window was 78%.

Out of the 24 models for the 1-year prediction window, one model reached a perfect PPV, while 11 models reached a 100% specificity. For the 2-year prediction window, 7 out of the 24 models reached a PPV of 100% while 9 reached a specificity of 100%. Overall, models based on all features had higher PPV. Model sensitivity and specificity had the best tradeoff when GBM was used. The highest model sensitivity was achieved using both DT and GBM, while the best specificity was achieved using RF, SVM, and XGBoost. When we compared the 3-year prediction window with and without the 24 days cut-off, the average AUROC, sensitivity, and specificity were unaffected; however, the average model accuracy was reduced by 5% when excluding the 24 days interval. Detailed performance metrics for the 288 models are presented in Appendix A.

### 3.3. Age, BMI, and Laboratory Values Highly Associated with Stroke Recurrence

Age and BMI had the highest overall feature importance at 90% ± 5% and 58 ± 10%, respectively. Laboratory values specifically LDL, HDL, platelets, hemoglobin A1c, creatinine, white blood cell, and hemoglobin were highly ranked in our different modeling frameworks. The feature importance of laboratory-based features ranged from 49% ± 10% to 39% ± 11% for HDL and platelet, respectively. Laboratory values had an average feature importance score of 44%, the highest among the different feature categories. Medications (statin, antihypertensive, warfarin, and antiplatelet), were also important features. Figure 4 (and Appendix A) includes the feature importance of different models and the overall average feature importance across the models and different prediction windows. The difference in days between the last outpatient visit before the index date and index date (45% ± 12%) and certain comorbidities were other important features for the recurrence models.

### 3.4. Models’ Performance Metrics Improved through Sampling Strategies

Given the low prevalence of recurrent stroke in our dataset (11–21%), we applied up- and down-sampling to the training dataset for the prediction window prior to the model training.

The application of up-sampling the minority class using 1:2 and 1:1 ratio for the 1-year prediction window improved the sensitivity to 55% while only slightly affecting the specificity to 91%. The model AUROC averaged 0.67 before up-sampling to 0.68 after up-sampling with five of the models reaching an AUROC above 0.75. The AUROC of the test set for the 3-year prediction window remained at 0.69 while the AUROC of the training set improved as expected with up-sampling (Figure 5, Appendix A).

## 4. Discussion

We have taken a comprehensive approach to develop and optimize interpretable models of long-term stroke recurrence. We have shown that (1) the six algorithms used could be trained to predict the long-term stroke recurrence, (2) many of the clinical features that were highly associated with stroke recurrence could be actionable, and (3) model performance metrics could be optimized.

There have been multiple clinical scores developed for predicting recurrence after cerebral ischemia with limited clinical utility [6]. Among all, only Stroke Prognostic Instrument (SPI-II) [26] and Essen Stroke Risk Score (ESRS) [27] were designed to predict the long-term (up to 2-years) risk of recurrence after an ischemic stroke. SPI-II can be applied to patients with transient ischemic attack (TIA) and minor strokes; yet, ESRS application focuses on stroke. The main limitations of SPI-II are focusing on patients with suspected carotid TIA or minor stroke, developed using a cohort of 142 patients. The ESRS, derived from the stroke subgroup of the clopidogrel versus aspirin among patients at risk of ischemic events (CAPRIE) trial, includes only eight parameters. In a validation study, the PPV for each tool were low, raising questions about their utility [28,29,30]. Previous validation studies of SPI-II demonstrated a *c*-statistic of 0.62 to 0.65, which can be judged as only fair [26,31,32]. In addition, SPI-II has poor performance in stratifying recurrent stroke in isolation as compared with the composite of recurrent stroke and death. The above demonstrates that the SPI-II score’s performance is driven mostly by its ability to predict mortality not a recurrence. There is an unmet need for better predictive measures of long-term prediction given the high rate and devastating consequences of a recurrent stroke. Other studies over the past few years have shown the power of ML in predicting short and long-term outcomes in various complex diseases [8,9,25].

### 4.1. Models Could Be Trained to Predict the Long-Term Stroke Recurrence

Our results showed that a high-quality training dataset with a rich set of variables can be utilized to develop models of recurrent stroke. Among the 288 models, prediction of stroke recurrence within a 1-year prediction window had an AUROC of 0.79, an accuracy of 88% (95% CI: 84%–91%), PPV of 42%, and specificity of 96% using RF with up-sampling the training dataset (Appendix A, model number 63). The LR-based models have similar results when compared to more complex algorithms such as XGBoost or RF. Our results showed that 21 (7%) models reached an AUROC above 0.73 while 110 (38%) models reached an AUROC above 0.7. Furthermore, the AUROC for the training and testing dataset were within a similar range which corroborates that models were not suffering from over-fitting. As expected, a model based on LR took a fraction of the time for training when compared to XGBoost, RF, or SVM (Appendix A).

We tested the prediction window for up to 5-years. Our results showed that the average model accuracy declined from 85% for the 1-year window to 78% for the 5-year window. However, the shorter prediction window provided the lowest rate of recurrence and therefore highest data imbalance, affecting model performance. The average model sensitivity increased as the prediction window widened, likely due to the increase in sample size and recurrent stroke rate. The optimal prediction window could depend on the richness of longitudinal data used for model training, in our dataset, that was between 2 and 4-years.

### 4.2. Clinical Features Highly Associated with Stroke Recurrence

In this study, 53 features were used as the full set (set1), followed by a subset of features excluding medication history (Set 2, 31 features). We also applied feature selection and created data-driven features (Set 3) and a minimum set of features (Set 4) for comparison. In most of the experiments more comprehensive feature set led to higher model performance, even though some features had some level of collinearity. In general, baseline clinical features, such as age, BMI, and laboratory values were among the most important features. Our results also highlighted that the last outpatient visit before the index stroke was important for the prediction of recurrence; patients in the control group had the lowest average number of days when compared to the five different case groups.

Analyzing the feature importance revealed that in general laboratory values were highly influential in the prediction models. The pattern of the importance of features was similar when considering different prediction windows, with many comorbidities and medications having the lowest relative impact. Laboratory values (LDL, HDL, platelets, HbA1c, creatinine, and hemoglobin), and blood pressure have shown to be high-ranking for all of the five different prediction windows and all of the different modeling framework with few exceptions. This finding highlights the fact that these potentially actionable features (e.g., HbA1c) may have more importance when compared to the corresponding comorbidities in the patient’s chart. The binary nature of medical history without the corresponding measures may have limited power in predicting recurrence. However, one of the main limitations of using more comprehensive laboratory values is missingness, especially when the missing is not completely at random.

### 4.3. Model Performance Metrics Optimized Based on the Target Goals

We have also shown that model performance metrics, such as specificity and sensitivity can be optimized based on the availability of resources and institutional priorities. We were able to improve the sensitivity of the models for the 1 and 3-year prediction window by sampling the training dataset to address the data imbalance. The tradeoff between specificity and sensitivity was of special interest given that different healthcare systems likely have different constraints, availability of resources, and infrastructures to implement preventive strategies to reduce stroke burden. Some of the resources may include, the number of providers needed to schedule follow up appointments or to discuss medication plans and ensure that the patient is compliant; or availability of resources to provide home-care or telehealth for patients needing those services for continuity of care. Thus, optimizing sensitivity and specificity should be aligned with the institution’s priorities. Here we demonstrated that sampling strategies could be useful tools in achieving optimal tradeoffs by increasing the sensitivity of the models up to 55% even with a low rate of stroke recurrence.

### 4.4. Study Strengths, Limitations, and Future Directions

The EHR data used in model development was longitudinally rich. However, that also leads to some of the study limitations. There is an inherent noise associated with the use of administrative datasets such as EHR, including biased patient selection and lack of information regarding stroke severity captured for approximately half of the patients. However, separate logistic regression models were employed to study the association of NIHSS with one-year stroke recurrence and did not show any association (OR: 1.01, 95% CI: 0.97–1.05, *p* = 0.625). Our phenotype definition to extract patients with stroke was strict, leading to 100% specificity on a randomly selected sample, which also means that our criteria likely missed some of the cases (for instance, if the patient had some MRI contraindication). Nevertheless, MRI is part of our stroke order-set and is performed for every stroke patient unless the patient refuses or has a contraindication (e.g., non-compatible pacemaker, etc.). We also did not include transient ischemic attacks since it is associated with significant misdiagnosis [33].

As future directions, we are expanding this study at two different levels by including additional layers of data and improving the model and model optimization. We are expanding the GNSIS dataset by incorporating a larger number of laboratory-based features; unstructured data from clinical notes such as signs and symptoms during the initial phases of patient evaluation; information about stroke subtypes; and genetic information from a subset of patients enrolled in the MyCode initiative [34]. We are also expanding our modeling strategies by (1) improving the imputation for laboratory values for EHR-mining [35,36], which could improve patient representation and reduce algorithmic bias; (2) applying natural language processing to expand the feature set from clinical notes; (3) developing polygenic risk score [37] using genetic information from a subset of our GNSIS cohort; (4) improving model parameter optimization using sensitivity analysis (SA)-based approaches [38,39,40,41]; and (5) expanding the study by incorporating more advanced methodologies, including deep learning models to compare with binary classification developed in this study. Finally, we are planning on developing models that account for the competing risk of death and other major vascular events in addition to ischemic stroke.

In conclusion, predicting long term stroke recurrence is an unmet need with high clinical impact for improved outcomes. Using rich longitudinal data from EHR and optimized ML models, we have been able to develop models of stroke recurrence for different prediction windows. Model performance metrics could be optimized and implemented in the same healthcare system as an intelligent decision support system to improve outcomes. Even though validating the model in patients recruited at a later time point could be done within the Geisinger system, external validation will be necessary to predict how the model predictions may be affected with regard to other health care systems and patient demographics. External validation to assess generalizability and identify potential biases will be an important next step of this study as well. Finally, based on our findings, we recommend that studies aimed at using ML for the prediction of stroke recurrence should leverage more than one modeling framework, ideally including also logistic regression as benchmarking framework for comparison.

## Figures and Tables

**Figure 1 jcm-10-01286-f001:**
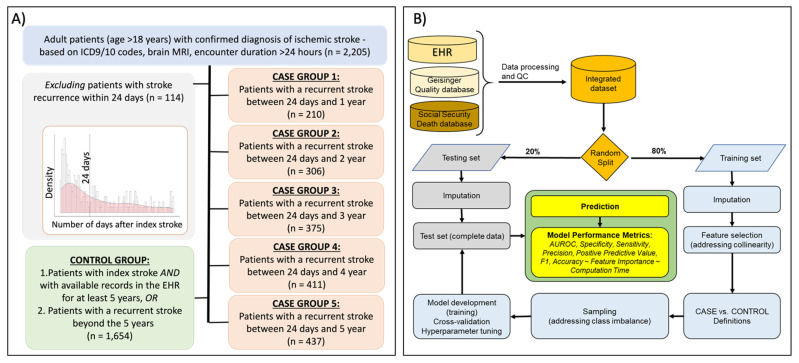
(**A**) Flow-chart of inclusion-exclusion of subjects in cases and control group in the study. Patients in the control group had available records in the electronic health record for at least 5 years and no documented stroke recurrence within 5 years. Distribution panel shows the number of recurrences over time. At 24 days, the number of recurrent cases can be seen to approach a plateau. (**B**) The design strategy for predicting stroke recurrence using electronic health records (EHR), Geisinger Quality database as well as Social Security Death database.

**Figure 2 jcm-10-01286-f002:**
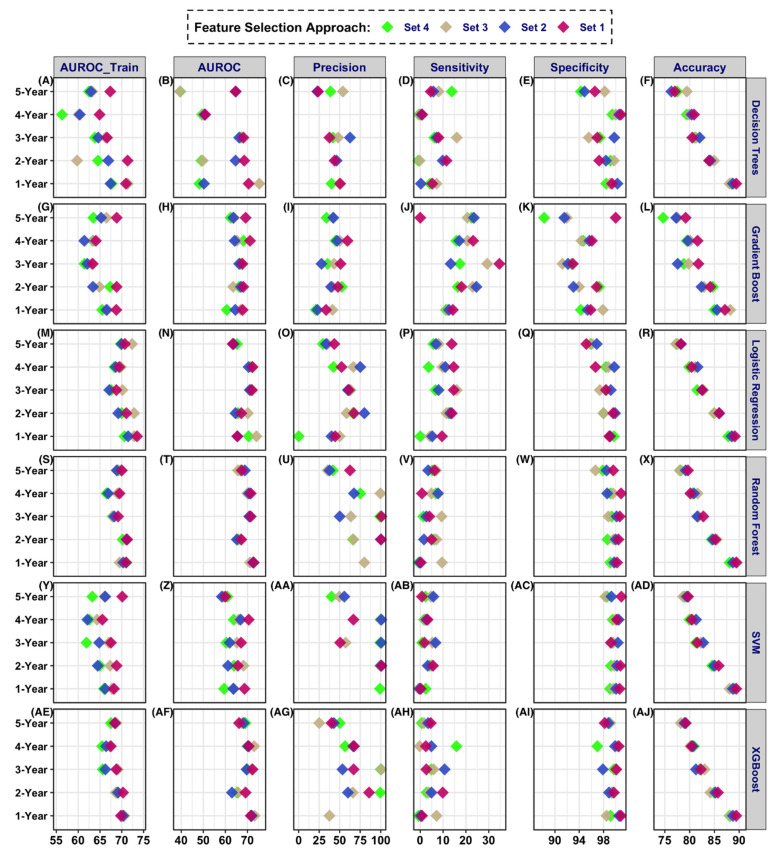
Model performance summaries for the five different prediction windows, six different classifiers, and four feature selection approaches. Performance metrics for (**A**–**F**) Decision tree, (**G**–**L**) Gradient Boost, (**M**–**R**) Logistic Regression, (**S**–**X**) Random Forest, (**Y**–**AD**) SVM, and (**AE**–**AJ**) XGBoost.

**Figure 3 jcm-10-01286-f003:**
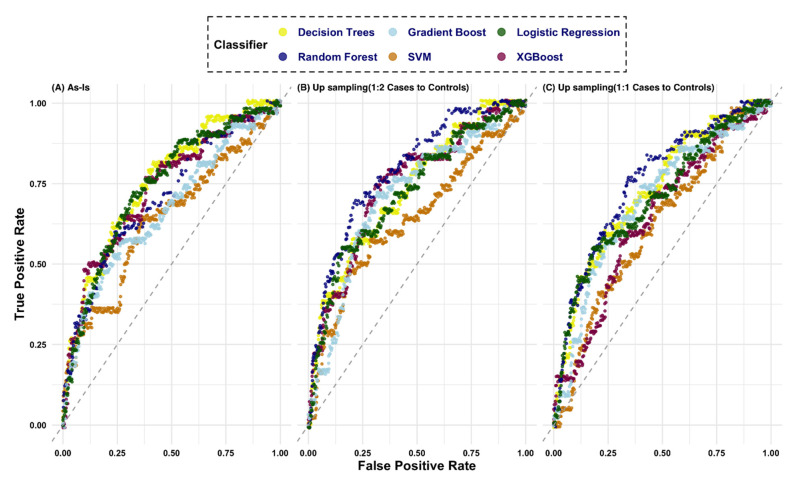
Area under the receiver operating characteristic (AROC) curve using six classifiers for the 1-year prediction window. The feature Set 3 is used for this figure. (**A**) Model without sampling; (**B**) Model with up-sampling at a 1:2 ratio; (**C**) Model with up-sampling at a 1:1 ratio. The best performer model (AUROC of 0.79) is when up-sampling is used with Random Forest algorithm (panel B).

**Figure 4 jcm-10-01286-f004:**
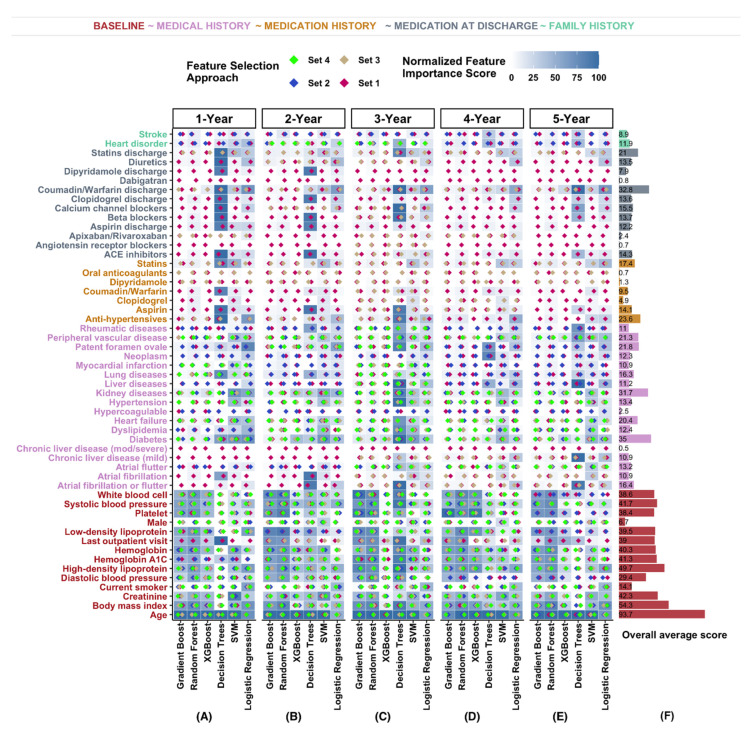
Feature importance based on the different trained models. (**A**–**E**) Six different classifiers (Gradient Boost, Random Forest, Extreme Gradient Boosting (XGBoost), Decision Trees, Support Vector Machine (SVM), and Logistic Regression) and five different prediction windows were used. (**F**) Average feature importance score across the different models and prediction windows.

**Figure 5 jcm-10-01286-f005:**
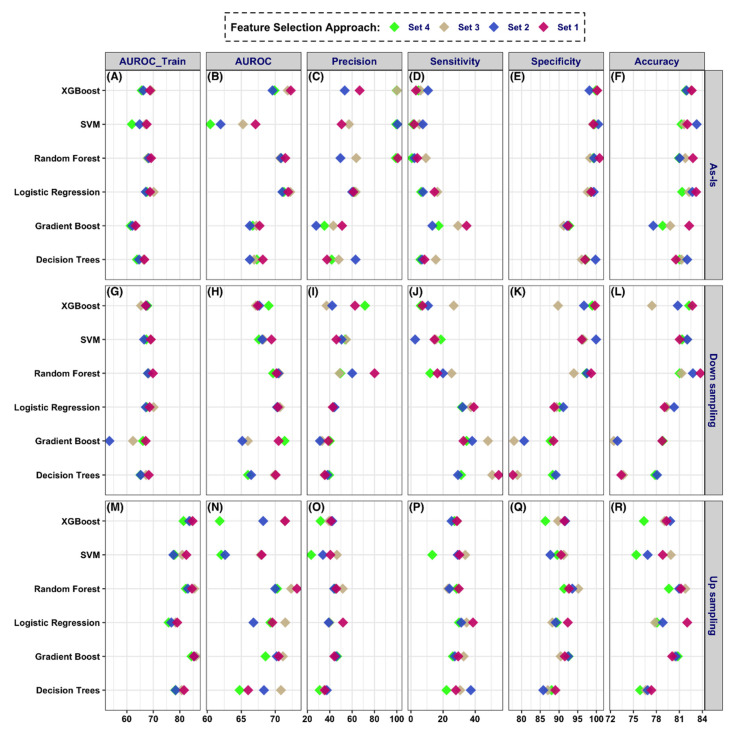
Model Performance summaries with sampling-based optimization for the 1 and 3-year prediction window. Up-sampling using was performed using the Synthetic Minority Over-sampling Technique (SMOTE). The feature Set 3 is used for this figure. (**A**–**F**) Model without sampling; (**G**–**L**) Model with down-sampling; (**M**–**R**) Model with up-sampling.

**Table 1 jcm-10-01286-t001:** Patient demographics, past medical and family history in different groups. Detailed description of the variables is provided in the Geisinger Neuroscience Ischemic Stroke (GNSIS) study [13]. IQR: interquartile range; HDL: high-density lipoprotein; LDL: low-density lipoprotein.

Patient Characteristics	% Missing	Statistics (All Patients)	Control Group	Case Group 1	Case Group 2	Case Group 3	Case Group 4	Case Group 5
Total number of patients	-	2091	1654	210	306	375	411	437
Age in years, mean (SD)	-	67 (13)	66 (13)	71 (14)	71 (13)	71 (13)	71 (13)	71 (13)
Age in years, median (IQR)	-	68 (58–77)	67 (57–76)	73 (62–83)	72 (63–81)	73 (63–81)	73 (63–81)	73 (63–81)
Male, n (%)	-	1079 (52%)	53%	47%	46%	46%	47%	47%
Body mass index (BMI) in kg/m^2^, mean (SD)	2.63%	30 (7)	30 (7)	29 (6)	29 (6)	29 (6)	29 (7)	29 (6)
Body mass index (BMI) in kg/m^2^, median [IQR]	2.63%	29 (26–33)	29 (26–33)	28 (24–32)	28 (25–32)	28 (25–32)	28 (25–32)	28 (25–32)
Diastolic Blood Pressure, mean (SD)	31.90%	76 (12)	76 (12)	75 (13)	75 (12)	75 (12)	75 (12)	74 (12)
Systolic Blood Pressure, mean (SD)	31.90%	137 (22)	136 (22)	139 (26)	139 (25)	140 (24)	139 (24)	139 (24)
Hemoglobin (Unit: g/dL), mean (SD)	1.82%	14 (2)	14 (2)	13 (2)	14 (2)	14 (2)	14 (2)	14 (2)
Hemoglobin A1c (Unit: %), mean (SD)	25.11%	7 (2)	7 (2)	7 (2)	7 (2)	7 (2)	7 (2)	7 (2)
HDL (Unit: mg/dL), mean (SD)	5.40%	47 (15)	47 (15)	45 (13)	45 (14)	45 (14)	45 (14)	45 (14)
LDL (Unit: mg/dL), mean (SD)	5.79%	102 (40)	103 (40)	103 (44)	100 (43)	101 (42)	101 (41)	100 (41)
Platelet (Unit: 10^3^/uL), mean (SD)	1.82%	232 (77)	233 (76)	227 (70)	229 (73)	231 (80)	230 (78)	229 (78)
White blood cell (Unit: 10^3^/uL), mean (SD)	1.82%	9 (3)	9 (3)	8 (3)	8 (3)	9 (3)	9 (3)	9 (3)
Creatinine (Unit: mg/dL), mean (SD)	2.58%	1 (1)	1 (0.5)	1 (1)	1 (1)	1 (1)	1 (1)	1 (1)
Current smoker, n (%)	-	288 (14%)	14 (1)	12 (6)	12 (4)	13 (3)	13 (3)	13 (3)
Difference in days between Last outpatient visit prior to index date and index date, mean (SD)	26.16%	347 (726)	345 (691)	371 (882)	354 (846)	369 (855)	352 (826)	354 (840)
MEDICAL HISTORY, n (%)								
Atrial flutter		41 (2%)	28 (2%)	4 (2%)	9 (3%)	11 (3%)	13 (3%)	13 (3%)
Atrial fibrillation		319 (15%)	230 (14%)	35 (17%)	55 (18%)	72 (19%)	82 (20%)	89 (20%)
Atrial fibrillation/flutter		324 (15%)	233 (14%)	36 (17%)	56 (18%)	74 (20%)	84 (20%)	91 (21%)
Chronic Heart failure (CHF)		159 (8%)	103 (6%)	33 (16%)	42 (14%)	49 (13%)	53 (13%)	56 (13%)
Chronic kidney disease		223 (11%)	142 (9%)	55 (26%)	68 (22%)	74 (20%)	78 (19%)	81 (19%)
Chronic liver disease		35 (2%)	23 (1%)	2 (1%)	7 (2%)	10 (3%)	11 (3%)	12 (3%)
Chronic liver disease (mild)		33 (2%)	21 (1%)	2 (1%)	7 (2%)	10 (3%)	11 (3%)	12 (3%)
Chronic liver disease (moderate to severe)		7 (0.3%)	5 (0.3%)	0 (0%)	1 (0.3%)	1 (0.3%)	2 (0.5%)	2 (0.5%)
Chronic lung diseases		391 (19%)	296 (18%)	51 (24%)	70 (23%)	83 (22%)	92 (22%)	95 (22%)
Diabetes		615 (29%)	439 (27%)	86 (41%)	122 (40%)	151 (40%)	165 (40%)	176 (40%)
Dyslipidemia		1298 (62%)	994 (60%)	142 (68%)	211 (69%)	258 (69%)	285 (69%)	304 (70%)
Hypertension		1495 (72%)	1150 (70%)	168 (80%)	240 (78%)	293 (78%)	327 (80%)	345 (79%)
Myocardial infarction		215 (10%)	159 (10%)	30 (14%)	43 (14%)	51 (14%)	53 (13%)	56 (13%)
Neoplasm		284 (14%)	211 (13%)	35 (17%)	49 (16%)	61 (16%)	65 (16%)	73 (17%)
Hypercoagulable		29 (1%)	24 (1%)	4 (2%)	4 (1%)	5 (1%)	5 (1%)	5 (1%)
Peripheral vascular disease		313 (15%)	219 (13%)	46 (22%)	65 (21%)	75 (20%)	88 (21%)	94 (22%)
Patent Foramen Ovale		241 (12%)	184 (11%)	30 (14%)	41 (13%)	47 (13%)	53 (13%)	57 (13%)
Rheumatic diseases		76 (4%)	53 (3%)	11 (5%)	14 (5%)	18 (5%)	21 (5%)	23 (5%)
FAMILY HISTORY								
Heart disorder		943 (45%)	747 (45%)	85 (40%)	130 (42%)	165 (44%)	182 (44%)	196 (45%)
Stroke		361 (17%)	279 (17%)	39 (19%)	60 (20%)	72 (19%)	77 (19%)	82 (19%)

## Data Availability

The data analyzed in this study are not publicly available due to privacy and security concerns. The data may be shared with a third party upon execution of data sharing agreement for reasonable requests; such requests should be addressed to Vida Abedi or Ramin Zand. Codes and additional meta-data, summary plots, and information can be found at https://github.com/TheDecodeLab/GNSIS_v1.0/tree/master/ModelingStrokeRecurrence (accessed on 19 March 2021).

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
