# Peer review of "Prediction of Long-Term Stroke Recurrence Using Machine Learning Models"

_jcm, 2021, doi:10.3390/jcm10061286_

Round 1

Reviewer 1 Report

Dear authors,

In this paper, the authors reported the prediction performance of several machine learning classifiers to predict the stroke recurrences in patients with ischemic stroke. The experimental results on “Geisinger NeuroScience Ischemic Stroke (GNSIS) database were promising. Overall, the manuscript is well written and easy to understand. However, it exits some limitations that should be addressed for further improving the manuscript.

<Major point>

  1. In “inclusion criteria”, the authors divided the patients into 6 groups: the control group and case 1, 2, 3, 4, 5 groups. Especially, patients with stroke-related or other vascular death might be excluded in this study. Therefore, the importance of stroke recurrence may be underestimated in this study. More accurate information about how patients with stroke-related death were allocated can help readers understand more properly.

  1. In my opinion, presenting Harrel's C-index using the Cox-proportional hazards model, survival regression, or random forest survival would show much clearer results compared to binary classifiers in this stroke recurrence study. In this paper, too many algorithms and tuning experiments are diluting the importance of the study results. In this regard, it would be better to add the need for survival ML or deep learning model as a limitation or future perspectives.

<Minor point>

  1. Please describe in the footnote to help readers understand that the values presented in Table 1 were measured at the time of index stroke.

Author Response

In this paper, the authors reported the prediction performance of several machine learning classifiers to predict the stroke recurrences in patients with ischemic stroke. The experimental results on “Geisinger NeuroScience Ischemic Stroke (GNSIS) database were promising. Overall, the manuscript is well written and easy to understand. However, it exits some limitations that should be addressed for further improving the manuscript.

- Thank you for reviewing our manuscript and providing constructive feedbacks and suggestions.

<Major point>

  1. In “inclusion criteria”, the authors divided the patients into 6 groups: the control group and case 1, 2, 3, 4, 5 groups. Especially, patients with stroke-related or other vascular death might be excluded in this study. Therefore, the importance of stroke recurrence may be underestimated in this study. More accurate information about how patients with stroke-related death were allocated can help readers understand more properly.

    - Thank you for pointing this out. Patients with stroke-related or other vascular death are excluded based on the inclusion/exclusion criteria. We will add this information to the methodology to ensure the study is clear to the readers. Predicting composite outcome or mortality is important, however it is beyond the scope this study. We will also add a clarification in the study limitation about this observation. 
  2. In my opinion, presenting Harrel's C-index using the Cox-proportional hazards model, survival regression, or random forest survival would show much clearer results compared to binary classifiers in this stroke recurrence study. In this paper, too many algorithms and tuning experiments are diluting the importance of the study results. In this regard, it would be better to add the need for survival ML or deep learning model as a limitation or future perspectives.

    - Thank you for this comment and recommendations. This study is a first step towards our understanding and predicting long term stroke recurrence. As we have seen in our cohort (data not shown in this manuscript), our rate of stroke recurrence is steadily increasing while the rate of mortality post-stroke is decreasing. Therefore there is an unmet need to identify patients at risk of recurrence for more targeted secondary prevention. We are currently working on expanding the study by incorporating more advanced methodologies, including deep learning models, and incorporating data from a second healthcare system for assessing model generalizability. We will add this point into our future direction more clearly. 

<Minor point>

  1. Please describe in the footnote to help readers understand that the values presented in Table 1 were measured at the time of index stroke.

    - Thank you for this comment. We will add more clarification to the Table 1.

Reviewer 2 Report

This is a well written and well organized paper.  It flows well and the key points are well put.  A couple of issues I think should be addressed are as follows:

  1. Geisinger is located in central Pennsylvania.  It is unclear how generalizable the results are to the general population.  The authors should discuss this at least in the limitations.
  2. The authors present several methods and argue that any of the methods presented could improve the clinical risk scores used in practice.  However, they don't provide any guidance beyond reporting of performance outcomes and solution speed.  It would be beneficial to understand if there are important differences that should be considered for practice.  One could conclude (I believe falsely) that it doesn't really matter.

Author Response

This is a well written and well organized paper.  It flows well and the key points are well put.  A couple of issues I think should be addressed are as follows:

Thank you for reviewing our manuscript and providing constructive feedbacks. 

  1. Geisinger is located in central Pennsylvania.  It is unclear how generalizable the results are to the general population.  The authors should discuss this at least in the limitations.

    This point is correct, thank you for bringing this to our attention. We will add this limitation in the discussion. In fact, we are currently working on expanding the study by collaborating with a second  healthcare system for assessing model generalizability.

  2. The authors present several methods and argue that any of the methods presented could improve the clinical risk scores used in practice.  However, they don't provide any guidance beyond reporting of performance outcomes and solution speed.  It would be beneficial to understand if there are important differences that should be considered for practice.  One could conclude (I believe falsely) that it doesn't really matter.

    All the ML in this study are able to provide reasonable performance. However, our recommendation is to use at least few, including LR as benchmarking framework for comparison. We will add a concluding statement. However, testing generalizability of the models remains an important point, and a more solid recommendations will be made once we have corroborated that these models are suitable based on external validation studies.